

# Using linear measurements to diagnose the ecological habitat of *Spinosaurus*

Sean Smart and Manabu Sakamoto

School of Biological Sciences, University of Reading, Reading, United Kingdom

## ABSTRACT

Much of the ecological discourse surrounding the polarising theropod *Spinosaurus* has centred on qualitative discussions. Using a quantitative multivariate data analytical approach on size-adjusted linear measurements of the skull, we examine patterns in skull shape across a range of sauropsid clades and three ecological realms (terrestrial, semi-aquatic, and aquatic). We utilise cluster analyses to identify emergent properties of the data which associate properties of skull shape with ecological realm occupancy. Results revealed terrestrial ecologies to be significantly distinct from both semi- and fully aquatic ecologies, the latter two were not significantly different. Spinosaurids (including *Spinosaurus*) plotted away from theropods in morphospace and close to both marine taxa and wading birds. The position of nares and the degree of rostral elongation had the greatest effect on categorisation. Comparisons of supervised (k-means) and unsupervised clustering demonstrated categorising taxa into three groups (ecological realms) was inappropriate and suggested instead that cluster division is based on morphological adaptations to feeding on aquatic *versus* terrestrial food items. The relative position of the nares in longirostrine taxa is associated with which skull bones are elongated. Rostral elongation is observed by either elongating the maxilla and the premaxilla or by elongating the maxilla only. This results in the nares positioned towards the orbits or towards the anterior end of the rostrum respectively, with implications on available feeding methods. Spinosaurids, especially *Spinosaurus*, show elongation in the maxilla-premaxilla complex, achieving similar functional outcomes to elongation of the premaxilla seen in birds, particularly large-bodied piscivorous taxa. Such a skull construction would bolster "stand-and-wait" predation of aquatic prey to a greater extent than serving other proposed feeding methods.

## INTRODUCTION

The enigmatic theropod *Spinosaurus aegyptiacus* (*Stromer, 1915*) is putatively considered 'semi-aquatic' to some capacity (*Aureliano et al., 2018*; *Henderson, 2018*; *Ibrahim et al., 2020*; *Fabbri et al., 2022a*; *Sereno et al., 2022*). This is supported by morphological (*Ibrahim et al., 2014*; *Beevor et al., 2021*), geographical (*Bertin, 2010*; *Benyoucef et al., 2015*), and isotopic (*Amiot et al., 2010*) evidence. Of specific interest are cranial adaptations to piscivory (itself indicative of aquatic affinities) observed in *Spinosaurus*; conical, interlocking dentition, anterodorsally elevated nares (distinct from posteriorly retracted

Corresponding author
Manabu Sakamoto,
m.sakamoto@reading.ac.uk

nares seen in cetaceans and other marine taxa (*Berta, Ekdale & Cranford, 2014*; *Hone & Holtz, 2021*) with implications for skull positioning in feeding (see discussion)), lateral skull compression, and raised orbit position (*Ibrahim et al., 2014*; *Arden et al., 2019*; *Hone & Holtz, 2021*). Partial piscivory has been widely advocated for across Spinosauridae (*Charig & Milner, 1997*; *Allain et al., 2012*; *Sales & Schultz, 2017*; *Fabbri et al., 2022a*), though only directly observed in *Baryonyx* (*Charig & Milner, 1997*). Spinosaurine spinosaurids show fewer, larger teeth with fluting in place of serrations compared to baryonychine spinosaurids, (*Sereno et al., 1998*; *Sales & Schultz, 2017*; *Hone & Holtz, 2021*). Larger fluted teeth have been proposed as adaptations to a diet including a larger proportion of hard-bodied prey (*Massare, 1987*; *Hone & Holtz, 2021*) in spinosaurine spinosaurids. Alternatively, this dentition could be an adaptation to withstand greater bite forces generated by greater body size compared to baryonychine spinosaurids (*Sakamoto, 2022*), similar selection pressures were suggested by *Sereno et al. (2022)*.

As described, *Spinosaurus* is considered to be semi-aquatic partially due to the shape of its skull and relative position of the orbits (*Ibrahim et al., 2014*; *Arden et al., 2019*; *Hone & Holtz, 2021*). Despite a fragmentary cranial fossil record, sufficient material of multiple spinosaurids (both those considered semi-aquatic and terrestrial) exists to compare these features and their relation to ecological realm occupancy, both within Spinosauridae and to other taxa where ecological realm occupancy is undisputed. To this end, cranial linear morphometric analyses (*Mosimann, 1970*; *Sakamoto & Ruta, 2012*; *Morales & Giannini, 2021*) have previously been successful in revealing the taxonomic affinities of unidentified specimens (*Blake et al., 2014*; *Naish et al., 2014*). Linear morphometric analysis could thus be a suitable method for categorising the ecological realm occupancy of *Spinosaurus*.

Within the *Spinosaurus* literature, definitions of 'semi-aquatic' can be varied. We define a 'semi-aquatic' animal to refer to those that utilise aquatic environments for a significant proportion of their nutritional resources and/or spend a significant proportion of time within bodies of water but retain terrestrial locomotory capabilities. However, a more specific diagnosis of the ecological realm utilisation of *Spinosaurus* beyond semi-aquatic would be advantageous, as there is no consensus thus far (*Ibrahim et al., 2014*; *Hone & Holtz, 2021*; *Fabbri et al., 2022a*; *Sereno et al., 2022*).

Most recent research broadly addresses one of two competing hypotheses: the 'underwater pursuit predator' hypothesis (*Ibrahim et al., 2020*) and the 'shallow water wading' hypothesis (*Hone & Holtz, 2021*). The former paints *Spinosaurus* as specialised in actively chasing down prey whilst submerged in the water column, propelled by tail and trunk (*Ibrahim et al., 2020*). In contrast, the latter hypothesis describes a hunting mode wherein the majority of the animal remains above the waterline, except for portions of the limbs and rostrum, suggesting stork-like feeding behaviours (*Paul, 1988*; *Hone & Holtz, 2017*).

The shallow water wading hypothesis has received notable support in recent publications (*Hone & Holtz, 2021*; *Sereno et al., 2022*), though discussions remain ongoing (see *Fabbri et al., 2022a*, *2022b*). Here, we expand on the work by *Hone & Holtz (2021)* to develop their quantitative approach to analysing cranial morphology in sauropsids across

different ecological realms. To this end, we aim to build upon the work of Hone and Holtz to evaluate whether ecological realm occupancy (terrestrial, semi-aquatic, or aquatic) can be inferred through skull morphometrics based on multiple linear measurements, and apply this to *Spinosaurus*.

## MATERIALS AND METHODS

A total of 99 taxa from eight clades were examined and subdivided by known or inferred ecologies (terrestrial, semi-aquatic (following the definition above), or aquatic).
Our dataset expanded upon that of *Hone & Holtz (2021)*, both increasing the number of representatives of previously examined clades and adding the families Ardeidae (herons) and Ciconiidae (storks), to allow morphological comparisons between *Spinosaurus* and these ecological analogues (wading birds) as proposed by *Hone & Holtz (2021)*. Taxa were selected upon the availability of photographs (*The Experimental Zoology Group of Wageningen University, 2022*), 3D scans (*The University of Texas at Austin, 2022*), or reconstructions depicting the skull in dorsal and lateral orientations (Supplemental Material).

For each taxon, six raw measurements (Fig. 1B) were taken using ImageJ v. 1.53 (*Abràmoff, Magalhães & Ram, 2004*) following *Hone & Holtz (2021)*. As some variable measurements were 0 (such as when naris lies on the skull anterior margin), to allow these values to be log-transformed, a constant of 1 mm was added to all measurements.
To account for isometric scaling due to body size, each measurement was divided by the geometric mean of the skull (*Sakamoto & Ruta, 2012*). The resultant dimensionless Mosimann shape variables have been demonstrated to out-perform residuals as size-adjusted shape variables, and have the additional benefit of only requiring information from a single specimen (*Mosimann, 1970*; *Sakamoto & Ruta, 2012*; *Morales & Giannini, 2021*). These values were log-transformed (*Glazier, 2013*), centred on 0 and scaled to unit variance to conform to the assumptions of cluster analysis and to a lesser extent, principal component analysis (PCA). Geometric mean was selected as a proxy for body size over skull length due to the presence of characteristic rostral elongation in multiple taxa examined (*Bertin, 2010*; *Erickson et al., 2012*; *Fischer et al., 2017*). Rostral elongation can interfere with body size estimates derived from skull length due to the allometric relationship between these variables (*Therrien & Henderson, 2007*).

### Data analyses

All data and statistical analyses were conducted in PAST v. 4.03 (*Hammer, Harper & Ryan, 2001*). A principal component analysis (PCA) was performed on the six Mosimann shape variables of 95 taxa (those with incomplete information were excluded) using a variance-covariance matrix. Taxa were grouped by clade and ecology. Permutational multivariate analyses of variance (PerMANOVA) were used to assess the overlap in morphospace between clades, and between ecologies. These analyses used a Euclidian similarity index and Bonferroni-corrected *p* values. To determine the appropriate number and pattern of clustering in morphospace, we used both classical (unsupervised clustering

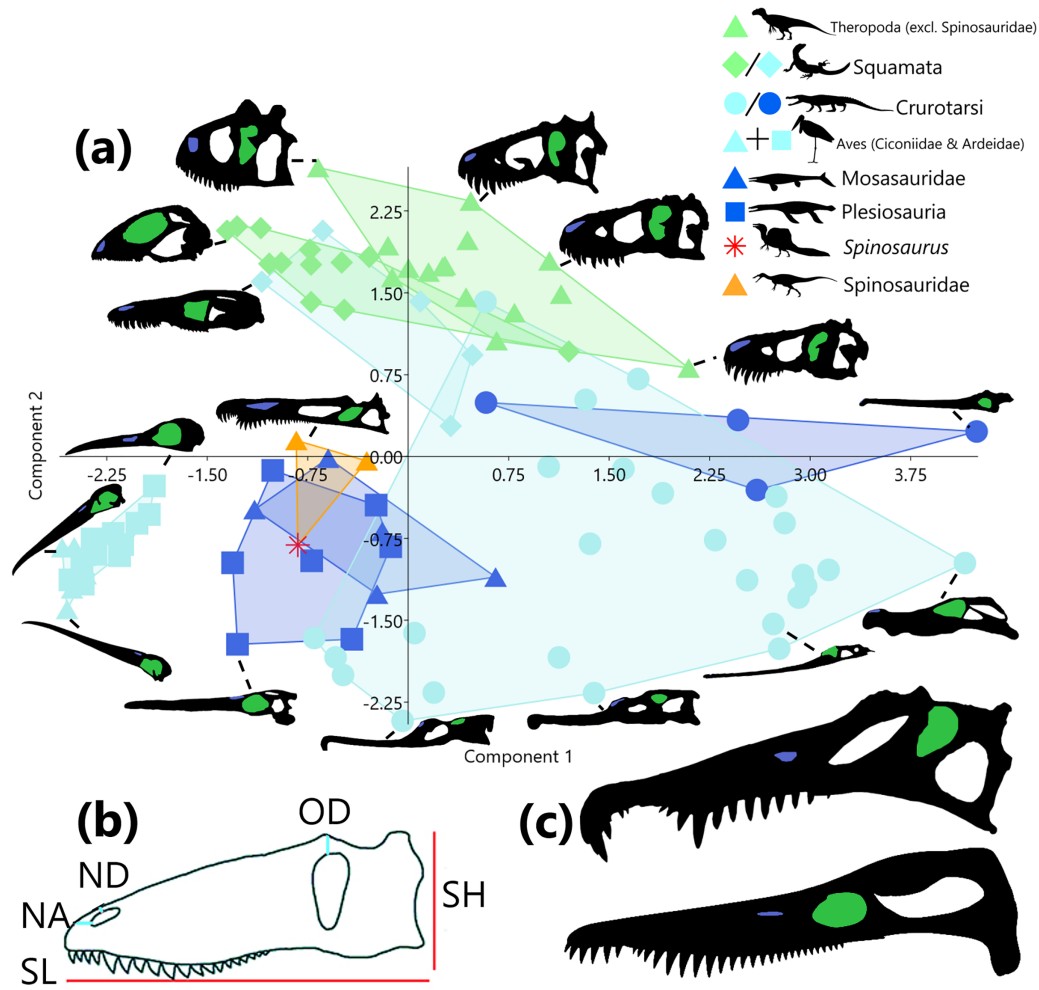

**Figure 1 Principal component analysis of six size-adjusted linear (Mosimann) variables in 95 representative taxa across 8 sauropsid clades (Squamata and Crurotarsi further divided based on ecology).** Skull silhouettes modified from respective sources (see Supplemental Material). (A) Distribution of taxa in morphospace of principle component (PC) 1 (48.06% of variance) and PC2 (30.10% of variance). Convex hulls delimit K-means (k = 3) cluster groupings. Green markers indicate terrestrial ecology, light blue markers indicate a semi-aquatic ecology, dark blue markers represent a fully aquatic ecology, and orange and red represent unknown ecologies. Silhouettes of taxa highlight the naris (blue) and the orbit (green). (B) Measurements were taken from the skull in lateral view. Skull height (SH), skull length (SL), distance from naris to anterior margin (NA), distance from naris to dorsal margin (ND), and distance from orbit to dorsal margin (OD). Not pictured: skull width (taken from dorsal view). Modified from *Hone & Holtz (2021)*. (C) Comparison of skull morphology between *Spinosaurus sp.* (top, modified from *Ibrahim et al., 2014*) and *Pliosaurus kevani* (bottom, modified from *Benson et al., 2013*). *Allosaurus* and *Spinosaurus* by Tasman Dixon, *Varanus* by Steven Traver, *Goniopholis*, *Tylosaurus*, *Rhomaleosaurus*, and *Baryonyx* by Scott Hartman, and *Leptoptilos* by L. Shyamal. Taxa silhouettes source credit: PhyloPic.org, CC BY 4.0, https://creativecommons.org/licenses/by/4.0.

using Ward's method and Euclidian distance measure) and k-means (supervised) cluster analyses to examine the relative effect of ecology compared to phylogeny on skull shape. We chose k = 3 for k-means analysis to reflect the number of ecologies.

## RESULTS

Following Principal component analysis (PCA) of linear skull measurements, PC1 and PC2 cumulatively account for 78.16% of the variance in the data, PC1 explaining 48.06% and PC2 explaining 30.10%. A morphospace plot of PC1 against PC2 (Fig. 1A) reveals that an increase along PC1 is associated with an increase in skull size and a decrease in NA. Whereas increase along PC2 is associated with skull heights approximately equal to skull length, progressively deeper-set orbits (long OD) and nares (long OD), and increasing distance from the naris to the skull anterior margin (long NA) (Supplemental Material). Ecologies are scattered across morphospace, though an increase along PC1 is associated with increasing terrestriality and higher values of PC2 (both positive and negative) are associated with increased aquatic affinities. When clades contain representatives from multiple ecologies (squamates and crurotarsans), members remain close in morphospace despite their assigned ecologies (Fig. 1A). Wading birds, spinosaurids, mosasaurs and plesiosaurs occupy the same region of morphospace, with *Spinosaurus* being most similar to *Pliosaurus kevani* (Fig. 1C).

A one-way PerMANOVA by clade, reveals that all aquatic taxa and semi-aquatic crurotarsans did not occupy significantly different ($p > 0.05$) regions of morphospace. Spinosaurids (excluding *Spinosaurus*) did not show significant differences from any other clade. Storks and herons, terrestrial and semi-aquatic squamates, and semi-aquatic and aquatic crurotarsans could not be differentiated from each other. Non-spinosaurid theropods were distinct from all clades except semi-aquatic squamates and spinosaurids. Likewise, semi-aquatic crurotarsans were distinct from all non-aquatic, non-spinosaurid taxa. A one-way PerMANOVA of the same data grouped by ecology showed all terrestrial ecology pairs as significantly different ($p < 0.001$) but aquatic and semi-aquatic ecologies could not be differentiated.

Supervised clustering using k-means (k = 3) cluster analysis did not yield clusters according to the three ecological realms categories (Fig. 1A). Cluster 1 is characterised by narrow skulls, elongate rostra, and substantial nares to anterior margin distances (larger NA). Cluster 2 is also associated with rostral elongation but is distinguished from Cluster 1 by anterodorsally elevated nares (smaller OD and NA). Cluster 3 contains all other terrestrial taxa. High PC 1 scores indicate the absence of rostral elongation.

Unsupervised cluster analysis on the other hand produced two clusters (Fig. 2). Cluster A is formed of taxa with elongate skulls and long nares to anterior margin distances. Taxa in this cluster are all either aquatic (plesiosaurs and mosasaurs) or semi-aquatic (phytosaurs (Crurotarsi) and avians). Cluster B contains taxa with short nares to anterior margin distances, this includes all terrestrial taxa, semi-aquatic squamates, and non-phytosaur crurotarsans.

## DISCUSSION

The discrepancy between the number of clusters produced by supervised (3-Fig. 1A) and unsupervised (2-Fig. 2) cluster analyses demonstrates that the *a priori* categorisation (K = 3) based on the three ecological realms is not supported by our linear morphometric data. Cluster membership did not correspond to specified ecology (terrestrial, semi-
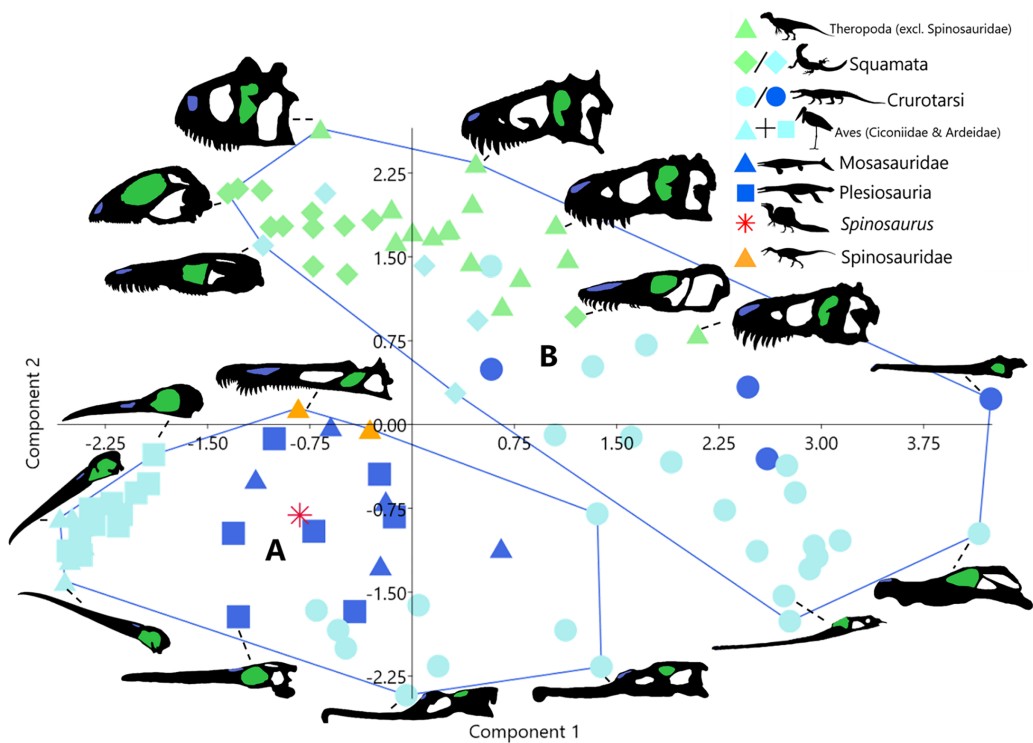

**Figure 2 Morphospace plot of principal component scores as in Fig. 1, showing instead unsupervised cluster analysis groupings.** Taxa silhouettes source credit: PhyloPic.org, CC BY 4.0, https://creativecommons.org/licenses/by/4.0.

aquatic, or aquatic) in either analysis. Taken together, this indicates that ecomorphologies associated with ecological realms are not emergent properties of our dataset. Instead, unsupervised cluster membership (Fig. 2) appears to be largely determined by the distance from the nares to the anterior margin (Supplemental Material), which dictates the relative rostral length often interpreted as adaptations for foraging in water, either semi- or fully submerged. Thus, separation in morphospace may be influenced by factors such as diet (feeding ecology, *i.e.*, what they are eating irrespective of ecological realm) and to a lesser extent phylogeny (*Melstrom et al., 2021*). This supports inferences regarding the evolutionary history of derived spinosaurids and feeding ecology in *Spinosaurus*.

Taxa in unsupervised cluster A (Fig. 2) exhibit proportionally posteriorly retracted nares relative to members of cluster B. Taxa in both clusters display rostral elongation, though this is achieved by elongation of different skull bones in each cluster. Members of cluster A (avians, plesiosaurs, mosasaurs, spinosaurids, and phytosaurs) extend the rostra *via* elongation of the skeletal elements rostral to the naris, *i.e.*, the maxilla-premaxilla complex. This distinguishes them from taxa in Cluster B (*i.e.*, non-phytosaurian crurotarsans), which elongate the rostrum *via* increasing the length of the nasal-maxilla complex (Supplemental Material). These two morphotypes attain the same outcome in terms of elongation of the rostrum (*e.g.*, increased reach), but differ in which part of the rostrum is elongated relative to the nares, anterior (Clade A) or posterior (Clade B).

The correlated movement of relative nares position results in significant impacts in which feeding modalities are available to these semi- and fully aquatic taxa.

In general, rostral elongation is viewed as an adaptation for both semi- and fully aquatic taxa feeding on aquatic prey. The associated increases in the out-lever distance (distance from jaw joint to bite point) results in greater relative jaw closing speed (*Sakamoto, 2010*; *Ballell et al., 2019*; *Evans et al., 2019*; *Brusatte et al., 2012*). This would be a desirable trait when feeding on highly mobile aquatic prey (*Massare, 1988*; *McCurry et al., 2017*) either fully submerged as in underwater pursuit predation or only submerging a portion of the skull as in the "stand-and-wait" strategy seen in herons and storks (*Kushlan, 1976*; *Willard, 1977*; *Maheswaran & Rahmani, 2002*).

Of these two main strategies proposed to describe feeding behaviour in *Spinosaurus* (*Ibrahim et al., 2020*; *Hone & Holtz, 2021*), the position of the nares is beneficial to the "stand-and-wait" strategy whilst being neither beneficial nor detrimental to the efficacy of the underwater pursuit strategy. An increased distance between the tip of the rostrum and the nares allows for a greater portion of the skull to remain submerged without restricting breathing, potentially increasing foraging efficiency (*Hone & Holtz, 2021*). In marine mosasaurs and pliosauroid plesiosaurs, rostral elongation is present but not as extreme, especially in plesiosauroid plesiosaurs (Supplemental Material). In these taxa, nares that are positioned closer to the dorsal margin of the head may have assisted with minimising the portion of the body exposed when surfacing for breath. Rostral elongation may also be less prominent in these taxa as jaw closing speed is associated with smaller prey items (*Morales-García et al., 2021*). There is strong evidence of niche partitioning in mosasaurs (*Madzia & Cau, 2020*; *Holwerda et al., 2023*) and niche conservatism associated with high bite forces in pliosaurid plesiosaurs (*Madzia & Cau, 2020*; *Foffa et al., 2014*) suggests they were not specialised for this specific type of prey. Minimising the proportion of the body exposed to the surface has also been suggested for instances when *Spinosaurus* is largely submerged and only extends a small portion of the head above the water, suggesting strong aquatic affinities (*Ibrahim et al., 2014*; *Arden et al., 2019*). However, the nares of *Spinosaurus* are not notably closer to the dorsal margin of the head than other spinosaurids for which such strong aquatic affinities are not proposed, and a larger portion of the head would have to be exposed, negating the proposed benefits (*Hone & Holtz, 2021*, *2022*). In contrast, the nares position of fully aquatic animals such as cetaceans and some marine reptiles, show a far greater movement towards the dorsal margin of the skull (*Hone & Holtz, 2021*).

Whilst it is difficult to address the sources of natural selection that differentiate the two evolutionary pathways to rostral elongation observed, as this was not directly tested, the relative lengths of the skull bones in *Spinosaurus* presents implications for the impacts of phylogeny on skull shape. Across all theropods examined (both spinosaurids and non-spinosaurids), the relative positions of the naris and orbit to the nasal remain constant (Supplemental Material). Interestingly, the rostral elongation in *Spinosaurus* is achieved *via* the lengthening of the skeletal elements anterior to the naris, thus maintaining the theropod configuration of the naris and orbit positions relative to the nasal. This feature is

more prominent in the derived *Spinosaurus* than in more basal baryonychine spinosaurids, which show more limited retraction (Figs. 1A, 1C). Avians display elongation primarily through elongation of the premaxilla as the maxilla is greatly reduced (*Bhullar et al., 2015*), piscivorous birds in particular having among the largest bills in terms of absolute size. In both birds and spinosaurids, elongation of the skull anterior to the nares serves to considerably increase the striking range, jaw closing speed, and amount of the skull that can be submerged whilst foraging, facilitating foraging on aquatic prey items.

In contrast, Crurotarsi displays both pathways of rostral elongation in phytosaurs (Cluster A) and crocodyliforms (Cluster B), demonstrating less phylogenetic constraint across Crurotarsi as a whole, but distinct effects of phylogeny within Phytosauria and Crocodyliformes respectively, *i.e.*, all phytosaurs are in Cluster A while all crocodyliforms are in Cluster B. The disparity in how rostral elongation is attained across Crurotarsi likely owes to it being a large and diverse clade, with phytosaurs being more basal, representing an older radiation, than crocodyliforms (*Nesbitt, 2011*). Regardless of what ecological selection pressures may be associated with rostral elongation, it is likely that members within clades attain this trait due to consistent selection pressures from feeding in aquatic environments. However, species engaging with such a feeding ecology are subjected to the mechanical pressures of feeding in a dense fluid medium that restricts viable morphospace (*Massare, 1988*; *Pierce, Angielczyk & Rayfield, 2008*). This combines with fewer modalities of feeding available to large-bodied marine organisms (*Taylor, 1987*) leading to convergence in skull shape and nares position (Fig. 2) though not necessarily convergence in diet.

The variables investigated here—in particular, the position of the nares relative to the anterior margin of the skull—are largely able to discriminate between taxa that feed terrestrially and those that feed on aquatic prey (Figs. 1A, 2). However, due to similar biomechanical restrictions, taxa that feed on aquatic prey show substantial skull shape convergence with fully aquatic taxa which prevents definitive categorisation of taxa with uncertain ecology (such as *Spinosaurus*), based on linear measurement of the skull alone. In regard to the feeding behaviour of *Spinosaurus*, elongation of the premaxilla in spinosaurids compared to other theropods results in a nares position which would have been exceedingly beneficial to the "stand-and-wait" predation strategy, but it is unclear to what extent it may be beneficial to the underwater pursuit predation strategy. Further investigations may consider post-cranial data, phylogenetic analyses and dentition, as well as feeding guilds as a categorisation factor. The data we have gathered is not sufficient to completely evaluate the utility of size-adjusted linear measurements of the cranium, but functions as an exploratory study which provides a framework for future studies to develop this line of enquiry. Specifically, we emphasise the importance of comparing supervised and unsupervised clustering to assess if the former is appropriate as the number of groups in supervised clustering can mislead inferences.

## ACKNOWLEDGEMENTS

Images used are available under the CC BY 4.0 licence: https://creativecommons.org/licenses/by/3.0/. Taxa silhouettes sources from PhyloPic.org.

### Funding
This research was funded by the University of Lincoln's Undergraduate Researcher Opportunity Scheme (UROS). The funders had no role in study design, data collection and analysis, decision to publish, or preparation of the manuscript.

### Grant Disclosures
The following grant information was disclosed by the authors:
University of Lincoln's Undergraduate Researcher Opportunity Scheme (UROS).

### Competing Interests
The authors declare that they have no competing interests.

### Author Contributions
- Sean Smart performed the experiments, analyzed the data, prepared figures and/or tables, authored or reviewed drafts of the article, and approved the final draft.
- Manabu Sakamoto conceived and designed the experiments, authored or reviewed drafts of the article, and approved the final draft.

### Data Availability
The raw measurements and Data for PAST software are available in the Supplemental Files.

### Supplemental Information
Supplemental information for this article can be found online at http://dx.doi.org/10.7717/peerj.17544#supplemental-information.

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
