# Peer review of "Using linear measurements to diagnose the ecological habitat of Spinosaurus"

_PeerJ, doi:10.7717/peerj.17544_

## Round 0.1 · original submission · Minor Revisions

Dear authors,

Based on the reviewers comments, I have made a decision of 'minor revisions'.

Reviewer one highlighted an important issue - what is the focus of the manuscript? There seems to be some confusion around what your hypotheses are, and thus what you can conclude. Can this be 'firmed up'? It would help with streamlining the text, and focusing on what your conclusions are.

Can you check reviewer two's comments re: figures. There seems to be an issue with the figures in the main text and the supplement.

I look forward to receiving your revised manuscript.

·

Basic reporting

This is basically fine, though there are various areas I have flagged up in the marked up copy that provide some areas that could be improved in clarity.

Experimental design

All fine, though it needs an additional bit of description as to how the ecological categories were applied. Although the authors have doubled the number of taxa in the analysis compared to that of Hone & Holtz, 2021, given the variety of taxa that have become secondarily aquatic or semi-aquatic, it would be informative to have a better spread here and a search of the literature would surely reveal at least a few more that could be measured and this would strengthen the results a lot.

Validity of the findings

Fine as far as they go, though the manuscript needs to be more clear about what it is investigating. It sort of flips between whether or not it is looking directly at the possible ecology of Spinosaurus or if it is assessing the part of the Hone & Holtz 2021 paper that used a very similar method to assess this. A clearer explanation of what the authors think they are assessing and then in the discussion what this means (see next section) would be useful.

Additional comments

I find that I'm not really sure what the authors think at the end of the paper. As per the point above, I'm not 100% sure what they think they are testing and therefore what the results mean. Given that they are diving into an issue that has been very contentious in recent years, real clarity is important here. What exactly has been assessed here and how do the results sit on the wading vs pursuit predator model discussion (and / or the issue of submergence / diving).

Reviewer 2 ·

Basic reporting

This paper extends previous research on the ecomorphology of Spinosaurus by expanding the database of taxa included and by teasing out the relative influence of particular morphologies loading on the different principal components.

Overall the paper is well written. It sets the question being studied in the context of the ongoing debates over spinosaurid ecology with reference to the relevant papers, and addresses some of the more ambiguous terminology (such as “semi-aquatic”) by clarifying the terms as they are employing it.

The most significant issue with the paper is the (at least to me) discordance between the description of the figures in the body text, the figures as provided in the body paper, and the versions of the figures as seen in the Supplementary data. As described in the abstract, Figure 1a is the morphospace plot of PC1 vs PC2 with the data labeled as their taxonomic component, 1b are the measurements taken, and 1c the comparison of the skull profiles of Spinosaurus and Pliosaurus: these are all consistent between the text and the caption.

However, in lines 116-120 there is a description of supervised clustering using k-means, resulting in three clusters labeled Cluster 1-3. This referenced as being in Figure 1a, but this matches Figure 2 in the body paper.

Furthermore Figure 2 as described in the text (showing the same data, but with unsupervised cluster analysis groupings) is said to yield only two clusters (A and B), but this does not match Figure 2 as provided (which distinctly shows three clusters labeled “1”, “2”, and “3”.) However, in the Supplementary Data on page 4 there is a figure which seems to conform to this description: a terrestrial cluster and a cluster made of the almost-entirely-overlapping sets of semi-aquatic and aquatic taxa.

My interpretation of the situation described above is that the Figure 2 in the body text really intended to be a separate figure showing the three supervised clustering, and a Figure 3 (the one in the Supplementary Data) showing the unsupervised clustering. The authors should please clarify this situation and relabel figures appropriately.

Experimental design

The paper is methodologically sound. It follows previous work but expands the taxa examined.

The data sources, methodologies followed, and software used are all provided or described as appropriate. Any other researcher should be able to duplicate the results if they so wished.

Validity of the findings

The conclusions of the authors are consistent with their analyses. They show that while there are morphometric similarities between Spinosaurus and some subaqueous feeders (such as Pliosaurus), but that the linear morphometrics as used here are not clearly discriminating between ecomorphology with regard to habitats.

Additional comments

Lines 77-78 In the caption for Figure 1, the sources for some but not all of the skull and body silhouettes are provided; the remaining sources (for instance, sources for the Carnotaurus and the tyrannosaurid skulls, among others) should be listed.

Line 138 Given that the other taxa (plesiosaurs, mosasaurs, etc.) are referred to in the vernacular, it might be more appropriate to refer to the members of Aves as either “avians” or just “birds”.

Lines 138ff In discussion of the morphologies, while it isn’t entirely inappropriate to refer to elongation of the “maxilla-premaxilla complex” in terms of herons and storks, this is really just the premaxillae in these cases. As with other neornithine birds, the maxilla is reduced to a tiny strut, and the rostrum is essentially only the premaxilla. (This condition is considerably different than even in the close relatives of neornithines, such as Ichthyornis or enantiornithines, much less any of the taxa in this study.)

---

## Round 0.2 · Minor Revisions

Dear authors,

Apologies for the delay in getting the review back to you. I have made a decision of 'minor revisions' based on the comments of the two reviewers.

I do not believe the comments from reviewer one will be hard to incorporate.

I look forward to receiving your revised manuscript.

·

Basic reporting

Generally good and much improved from the previous version. I do still have a few comments on the marked-up document where I think the clarity can be improved, especially with respect to the naris and the implications of its position.

Experimental design

No comment.

Validity of the findings

All good, and again clearer than before what was tested and the implications.

Additional comments

Some areas where some extra comments and explanations will help clarify things further and lead the reader through some complex areas, but these are few and minor in nature.

Reviewer 2 ·

Basic reporting

The earlier incarnation of this manuscript was already well-written, but I find the newer version of this even more clear.

The references are up-to-date, and the recent (often contentious) literature on the subject are present.

The raw data and the code used to study it are provided, so that others might duplicate the results. The structure of the paper is readable.

The study presented is narrowly-focused and is self-contained.

Experimental design

The methods and materials for the analysis are clearly indicated. The results presented seem coherent, and are interpreted in a conservative fashion. It elaborates on preliminary morphometric work, with greater scope both in terms of measurements chosen and taxa evaluated.

Validity of the findings

The conclusions presented are consistent with their analyses.

Additional comments

The changes from the earlier version of the manuscript are indeed improvements, and the paper is stronger for them.

---

## Round 0.3 · accepted · Accept

Dear authors,

Based on your response to reviewers .doc I have made the decision of ‘accept’. Thank you for constructively engaging with the peer-review process.

The production team will contact you to take you through the proofing stage.

Congratulations, and I hope you will use PeerJ as your publication venue again in the future.